



# Assessment of the effectiveness of Payment for Ecosystem Services (PES) in the delivery of desired Ecosystem Services in Sasumua catchment, Kenya

Charles Nduhiu[1], John Mwangi Gathenya[1], John Kimani Mwangi[2], Malik Aman[3] and Titus Mutisya[3]

[1]Department of Soil, Water and Environmental Engineering, Jomo Kenyatta University of Agriculture and Technology, Nairobi, Kenya
[2]Department of Civil Construction and Environmental Engineering, Jomo Kenyatta University of Agriculture and Technology, Nairobi, Kenya
[3]Kenya Agricultural Productivity and Sustainable Land Management Program, Ministry of Environment, Natural Resources and Regional Development Authorities Nairobi, Kenya

*Correspondence to:* Charles Nduhiu (nduhiucharles@gmail.com)

**Abstract.** The study was conducted in Sasumua watershed in Nyandarua County, Kenya where a Payment for Ecosystem Services (PES) pilot project was initiated in June 2015 with the aim of promoting sustainable land management practices (SLM) that would lead to improved water quality. This study which was conducted after one year of PES implementation, seeks to establish what effect the SLM technologies being promoted under PES would have on the water quality. A representative sub-watershed was established where 42.3 ha were under intensive cultivation. Baseline status on Total Suspended Solids (TSS) was established during the rainy season of March-May 2015 just before the onset of PES project. Baseline status on SLM technologies in the study site was also established. Two V-notches were installed to record flow in the rainy season of March-May 2016 for purposes of soil and water assessment tool (SWAT) calibration. Data collection involved water sampling at selected points during two major rainy seasons of October-December 2015 and March-May 2016. Water samples were tested for TSS by photometric determination method using Lovibond water quality testing kit. The SWAT model was applied to generate two scenarios (Worst and Best scenarios) of the study site. The scenarios before and after PES project (determined from actual field measurements) were fitted in between the SWAT scenarios to evaluate the effectiveness of the PES approach after one year of PES project. The baseline status for TSS was an average of 71.05 mg/L. After one year of PES project implementation, the TSS improved to an average of 42.73 mg/L. SWAT model predicts a worst scenario for TSS at an average of 124.15 mg/L and best scenario at an average of 12.76 mg/L. Watershed management through PES approach can be effective in improving downstream water quality as shown by increase in adoption of SLM technologies from 11% to approximately 32% within the first year. However, long term research data is highly recommended to validate the effectiveness of PES over number of years especially on ecosystem services that manifest after long periods and establishing whether PES incentives actually maintain best conditions at farm level. More ecosystem services should also be monitored to validate the TSS results.

**Key words**: Payment, Ecosystem Services, Sustainable Land Management, water quality,





## 1  Introduction

Over the past 50 years, humans have changed ecosystems more rapidly and extensively than in any comparable period of time in human history, largely to meet rapidly growing demands for food, fresh water, timber, fibre, and fuel etc (MEA, 2005) which has resulted in a substantial and largely irreversible loss in diversity of life on Earth.

Healthy watersheds provide valuable services to society, including supply and purification of fresh water. With population and development pressures leading to rapid modification of watersheds, valuable hydrological services are being lost which poses risks to quality of drinking water and reliability of water supplies (Sandra, *et al*, 2005). As watersheds determine water flows, they are appropriate areas for organizing planning and management of Ecosystem Services (ES) downstream (Smith, *et al* 2006).

Adoption of sustainable land management (SLM) technologies by upstream farmers can improve ecosystem services in form of improved water quality to downstream users. PES approach has been proven to provide desired ecosystem services (Porras *et al*., 2013; Pagiola, 2008). SLM practices are critical in reducing chemical and sediment pollution, improve rainwater retention, ground water recharge, regulate flows and wetland functions and reduce risk of floods and landslides. However, farmers are unable to adopt good land practices due to lack of

knowledge and financial resources. Several studies have reported slow uptake of sustainable land management practices through conventional approaches. (FAO, 2010) reported that adoption of SLM technologies has been relatively low globally. (World Bank, 2010) reported that adoption of SLM technologies in sub-Saharan Africa was very low-about 3% of total crop land. (Kihiu, 2016) estimated that in Kenya, the adoption rates of sustainable land management (SLM) practices in areas where SLM practices are highly needed (dry lands) due to unfavourable

conditions are alarmingly low estimated at 14.2%, despite the declining productivity of these ecosystems. Even though there is insufficient research done on adoption rates in all agro-climatic zones of Kenya, this value could be lower in semi-humid to humid zones as studies have reported that where lands are relatively productive, there is widespread apathy among small scale farmers to invest in SLM technologies as the perceived net gain is minimal (Sterve, 2010; Kirui, 2016; Molua, 2014).

Other studies report a generally low adoption or no adoption at all in Kenya through conventional approaches (Tanui *et al*., 2014; Branca *et al*., 2011; World Bank, 2008; Mulinge, *et al*., 2016; Shiferaw *et al*., 2009; Liniger *et al*., 2011; Jairo, 2013; MOA & MOE, 2011). Reward mechanisms such as PES that accelerate adoption rates through provision of incentives (Porras *et al*., 2013) are the alternative to the conventional approaches as incentives are highly likely to increase rate of SLM adoption by small scale farmers.

The Soil and Water Assessment Tool (SWAT) predicts the impact of land management practices on water, sediment, and agricultural chemical yields in watersheds with varying soils, land use, and management conditions over time (Arnold *et al*., 1998). The continuous-time, process-based model requires specific information about weather, soil properties, topography, vegetation, presence of ponds or reservoirs, groundwater, the main channel, and land management practices (Bracmort *et al* , 2006). SWAT model has been applied in several studies to quantify

effects of Sustainable Land management practices to water quality downstream (Cau & Paniconi, 2007; Gitau *et al*., 2008; Bracmort *et al*., 2006; Mbonimpa *et al*., 2012; Kieser, 2008).

Payments for environmental services (PES) are a class of economic instruments designed to provide incentives to land users to continue supplying an environmental service that is benefiting society more broadly. Payments may be made to land users to adopt land use practices that will produce the required service from scratch (e.g. planting grass

filters for buffering sediments loads). (Wunder Sven, 2005) defined PES as a voluntary transaction in which a well-defined environmental service (ES), or a form of land use likely to secure that service is bought by at least one ES buyer from a minimum of one ES provider if and only if the provider continues to supply that service.

Sasumua PES pilot project is a public scheme where the government represented by Kenya Agricultural Productivity and Sustainable Land Management Project (KAPSLM) is buying Ecosystem services as a dummy ES buyer. Table 1

shows chronology of KAPSLMP PES project implementation in Sasumua watershed. In June 2015, Kenya Agricultural and productivity and Sustainable Land Management Project (KAPSLMP) initiated a PES pilot project in Sasumua. This was to actualize '*theory into practice*' from past study findings and recommendations. KAPSLMP was acting as a dummy buyer of ES at the initial pilot stage. Prove of PES viability was required to enable replication and scaling up of the PES project. Therefore, assessing effectiveness of PES approach on soil erosion

control and water quality was an integral part of the KAPSLMP PES project. This study was established under the pilot project to assess the effectiveness of PES scheme in the delivery of desired ecosystem services in Sasumua catchment. The study focussed on two areas; (a) monitoring the progress in SLM adoption and (b) monitoring the





water quality in the strategically selected monitoring points in a way possible to correlate the SLM adoption to water quality improvement.

It was difficult to monitor more than 20,000 farmers in the larger Sasumua watershed given time and the resources available. This study established a small sub-watershed where it was practicable to follow the progress of every
farmer. The sub-watershed (headwater) is approximately 6.08km² with a total of 41 farmers engaged in PES pilot project. The selection was strategic such that all farmers had the same outlet point for monitoring reduction in sediments loads as they adopt new SLM practices. Furthermore, the choice of a headwater sub-basin ensured there were no interferences on the downstream results by upstream activities allowing a direct relationship of the water quality status downstream to the actual progress made on adoption of new SLM practices.

## 2    Materials and Methods
### 2.1 Site description
Sasumua is located in Central Kenya, southern ridges of Kenya's Aberdare Mountains about 90 km northwest of Nairobi, at an altitude of 2200-3850 m.a.s.l. Mean annual rainfall is 1000-1600 mm, peaking March-May and October-December in a binomial pattern (Gathenya *et al*, 2010). The catchment of the Sasumua reservoir is 107 km²
and comprises three sub-catchments: Sasumua (67.44 km²) Chania (20.23 km²) and Kiburu (19.30 km²). A representative sub-watershed (headwater) was established whereby 42.3 ha are under intensive cultivation. This was to make it possible to follow the actual actions of 41 households in the study site during the study period for deeper analysis on the effect of additional SLM technologies to downstream water quality. The headwater comprises of 3 sub-basins covering an area of 6.08 km². As shown in Fig 1, Forest is the dominant land cover for sub-basins 1 and 2
while agriculture is the dominant land cover in sub-basin 3. Table 7 shows characterization of the study site. There are a total of 67 farmers of which 41 are participating in the PES pilot project. The key point to note is that sub-division of land is high in this area, increasing the number of individuals who own land (some of whom are 'absent farmers[1]'). This number (67) is estimated from the farms with title deeds and not individuals. The actual number of individuals could be higher including the 'absent farmers[1]'.

### 2.2  Description of SWAT Model Inputs
The most important SWAT model inputs include the digital elevation model (DEM), land use and land cover, soil information and weather data. Table 3 describes input data information.

### 2.3  Representation of SLM practices in SWAT
For this study, four sustainable land management practices were considered including; terracing, contouring, filter
strips and strip cropping. These were based on key SLM practices being promoted by the PES pilot project in the Sasumua catchment (table 6).

A terrace is an embankment within the field constructed to intercept runoff and prevent erosion. Terracing in SWAT is simulated by adjusting both the run-off and erosion parameters. The USLE practice (TERR-P) factor, the slope length (TERR_SL), and curve number (TERR_CN) are adjusted to simulate the effects of terracing (Arnold *et al*.,
2012 and Waidler *et al*., 2011). Contour planting is a practice of tilling and planting along the contour of the field as opposed to straight row. Contour planting is simulated by altering the curve number (CONT_CN) to account for increased surface storage and infiltration and USLE practice (CONT_P) factor to account for decreased erosion (Arnold *et al*., 2012). A filter strip is a strip of dense vegetation located to intercept runoff from upslope sources and filter it. Filter strips are simulated by altering the ratio of field area to filter strip area (VFSRATIO), fraction of the
HRU (VFSCON) which drains to the most concentrated 10% of filter strip, and fraction of the flow (VFSCH within the most concentrated 10% of the filter strip which is fully channelized (Arnold *et al*., 2012). Strip cropping is the arrangement of bands of alternating crops within an agricultural field. Strip cropping is simulated by altering the Manning' N value for the overland flow (STRIP_N) to represent increased surface roughness in the direction of flow, curve number (STRIP_CN) to account for increased infiltration, USLE cropping (STRIP_C) factor to reflect
the average value of the multiple crops within the field and USLE practice factor (STRIP_P) to represent strip cropping conditions (Arnold *et al*., 2012).

---

[1] *'Absent farmers' - refers to individuals who own land but do not live in rural areas. They are participating in 'other' income generating activities typically in urban areas.*



### 2.4 Field data collection

Baseline status on SLM technologies was established at the onset of PES pilot project in June 2015 where 41 farmers were mapped and Land Management Plans (LMPs) established. These included: length in meters of grass filter strips planted, length in meters of terraces constructed and length in meters of protected riparian strip where applicable. Water samples were collected on a daily basis targeting the rainy seasons when soil erosion was expected to be high. Baseline status on TSS was carried out during the rainy season of March-May 2015 before the onset of PES pilot project. Subsequent measurements of TSS were carried out in two rainy seasons October-December 2015 and March-May 2016. Water samples were tested for Total Suspended Solids (TSS) by photometric determination method using a Lovibond water quality testing kit (Tintometer Group, Bibi *et al.*, 2011). The photometric method was also correlated with the conventional gravimetric method to determine the relationship. Two V-Notches were installed in February 2016 on which daily stream flows were observed in the rainy season of March-May 2016. The observed TSS and discharge were used in SWAT calibration. Daily rainfall data was collected from the existing weather station located in Sasumua catchment (Fig 7) and managed by Nairobi City Water and Sewerage Company (NWSC) that incorporates a standard rain gauge. Rainfall intensities were not measured in this study. Fig 6 shows that the wet season of March-May 2015 had a relatively higher rainfall. However, the numbers of storms with more than 30mm/day were relatively higher in the wet season of March-May 2016.

It should be noted that the measurement sites of TSS in sub-basins 1 and 2 (in fig 1) were not the same as the sites for which the discharge was calibrated. The distances between points A and C is approximately 356 meters and point B and D is approximately 286 meters. 96.7% of sub-basins 1 and 86.4% of sub-basin 2 are forest cover. In SWAT modelling, the HRU definition used a threshold of 15% in land use percentage which resulted in Sub-basin 1 and 2 being considered as forested lands and Sub-basin 3 as an agricultural land in SWAT model. The assumption therefore is that the TSS results at points A and B are insignificantly different from TSS at C and D given the small differences in catchment size between A & C, and B & D in comparison to the catchment sizes of sub-basins 1 and 2. Again validation of TSS used actual field data collected at the main outlet (Point E) in sub-basin 3. Validation of TSS was satisfactorily with a P-factor of 0.93 and NS of 0.70.

### 2.5 Model parameterization

After initial model run, it is necessary to parameterize the model to give it a realistic representation of the existing conditions in the watershed before model calibration. It is advisable to check simulation of the initial model set up and make sure simulations and observations are not too different as calibration will not work satisfactorily (Abbaspour, 2015).

For this study, the data on status of key SLM practices played a key role in model parameterization. After one year of PES project the status of SLM adoption had improved to 1561m of terraces, 551m of retention ditches, 3725m of grass strips, and 510m of river bank protection. Since there is no way to enter the lengths (metres) in the model, the values of the four SLM practices considered (terracing, contouring, filter strips and strip cropping) were entered based on interpretation of the (% status) and what that means in the SWAT parameter ranges.

For instance, the 1561m of terraces means 18% towards the target (table 4). One of the parameters to consider is TERR-P to reflect reduced sediment losses. The P-factor ranges between 0-1 in SWAT model. A value of 0 means 100% reduction and a value of 1 denotes 0% reduction. Therefore, a value of 0.82 was used for P-factor when status of terraces is recorded at 18% in the watershed. Other values considered during parameterization included the soil, crop and management.

### 2.6 Sensitivity analysis, model calibration and validation

SWAT input parameters are process based and must be held within a realistic uncertainty range. Parameters for sensitivity analysis were selected from literature and documentation from SWAT user manuals (Neitsch *et al.*, 2002; Arnold *et al.*, 2012). Firstly, the model was calibrated for flow before calibration of sediment parameters (Abbaspour, 2015). The flow data observed during the study period was used in calibration of discharge. The daily TSS observed during the same period was converted to metric tons for calibration purposes. Model calibration was done automatically using SUFI2 principle in SWAT-CUP (Abbaspour, 2015) using parameters identified as sensitive to discharge and sediments as shown in table 5.

Model calibration was considered satisfactory when the P-factor (percentage of the measured data bracketed by the 95PPU) was above 0.6 and when Nash-Sutcliffe (NS) was above 0.5 as recommended by (Bracmort *et al.*, 2006).





There were no long term observed data to allow temporal validation (i.e. division of data into calibration period and validation periods). However, spatial validation was adopted in this study. The TSS observed in sub-basin 3 was used in validating sediments simulation. The summation of discharges at sub-basins 1 and 2 was used as hypothetical discharge at sub-basin 3 for validation purposes. Hypothetical discharge was acceptable in this study as
the contributing area in sub-basin 3 was only 6.95% of the total area of the watershed.

### 2.7  Model runs

SWAT model simulations were performed to assess the effectiveness of SLM technologies to water quality downstream by specifically checking on the Total Suspended Solids (TSS) generated in two different scenarios (best and worst scenarios). The adjusted parameters (to reflect worst and best scenarios) were obtained from different
tables provided by (Arnold *et al.*, 2012) and judgement based on observation and field experience. For instance, appropriate USLE practice factor for well terraced field based on field slope are given in table 33-1 page 487 (Arnold *et al.*, 2012). Based on the slope of our study site, the appropriate P-factor is 0.18 as shown in table 6. To reprresent worst scenario, the P-factor was adjusted to 1 which reflects increased sediment losses (Arnold *et al*., 2012).

Best scenario is the scenario in which the selected SLM parameters were adjusted to their best values (i.e. when highly effective in reducing soil erosion), this scenario assumes all farmers adopt the recommended SLM technologies fully as stipulated in their LMPs and this situation will be maintained by the existence of PES incentives. Worst scenario is the scenario in which the selected SLM parameters were adjusted to their low values (i.e. not effective in reducing soil erosion) assuming poor state of SLM technologies in the study site. The scenarios
before and after PES project (established through actual field measurements) were fitted in between the SWAT generated scenarios (best and worst scenarios) to evaluate the effectiveness of PES approach (Fig 4).

### 2.8  Description of scenarios

#### 2.8.1 Scenario before PES-project (field observation)
This scenario was determined from actual field measurements on TSS during the rainy season of March - May 2015.
This was carried out before the onset of KAPSLMP PES project. The average reading during the three month rainy season was determined to represent conditions before PES project.

#### 2.8.2 Scenario after one year of PES project (field observation)
This scenario was also determined from actual field measurements on TSS during March - May 2016 rainy season, one year after PES pilot project. The average TSS was calculated to represent a scenario after one year of PES
project implementation.

#### 2.8.3 Best scenario (SWAT generated)
This scenario was determined by adjusting SWAT parameters for selected SLM technologies to their best considered status guided by literature (Arnold *et al.*, 2012) at which they are assumed to be highly effective in improving water quality downstream. All SWAT parameters are within certain ranges e.g TERR-P ranges between
35 0-1. A value approaching 1 reflects increased soil loss and a value approaching 0 reflects reduced soil loss (Arnold *et al.*, 2012).

#### 2.8.4 Worst scenario (SWAT generated)
This is where expert judgement played a key role based on observation and field experience which was further supported by interpretation of parameter ranges from literature. This scenario was represented by adjusting the
40 SWAT parameters to their worst considered status when the modelled SLM practices are assumed to be in poor state and not effective in improving water quality downstream. Table 6 shows selected SLM practices modelled in SWAT and the adjusted parameters.





## 3 Results and discussions

### 3.1 Calibration and validation - uncertainty analysis

Results indicated that calibration was satisfactory with P-factors above 0.6 and NS above 0.5. In flow calibration, the P-factor for sub-basin 1 was 0.76 and 0.71 for sub-basin 2. Nash-Sutcliffe (NS) for sub-basin 1 was 0.81 and
0.73 for sub-basin 2. In sediments calibration, the P-factor for sub-basin 1 was 0.71 and 0.69 for sub-basin 2. Nash-Sutcliffe (NS) for sub-basin 1 was 0.59 and 0.58 for sub-basin 2. Flow validation was satisfactory with P-factor of 0.67 and NS of 0.71 at the outlet of sub-basin 3. Sediments validation produced a P-factor of 0.93 and NS of 0.70. Figure 2 and 3 shows calibration and validation results developed in SWAT-CUP.

### 3.2 Impact of PES on water quality

TSS was used as the main proxy indicator in assessing the effectiveness of PES approach. The comparison of photometric method and the conventional gravimetric method in TSS measurements showed that the relationship is linear (Fig 8). Generally, the lovibond reads a relatively higher TSS value from that of gravimetric method with an average factor of 1.12. This means if conventional gravimetric method reads (X), the Lovibond reading can be estimated at (1.12X).

At the onset of PES pilot project in June 2015, the SLM baseline status included; 820m of terraces, 120m of retention ditches, 920m of grass strips, and 210m of river bank protection. The baseline status on TSS established during the rainy season of March – May 2015 was an average of 71.05 mg/L. After one year of PES project implementation, the SLM status improved to 1561m of terraces, 551m of retention ditches, 3725m of grass strips, and 510m of river bank protection. These figures were also used in model parameterization. At this period, the
average TSS observed during rainy season of March – May 2016 was an average of 42.73 mg/L.

Figure 5 shows the TSS results in the study site measured during two rain season of March-May 2015 and March-May 2016. The rainfall series in both seasons are as shown in figure 6. As noted earlier, the study site is a 'headwater', points A and B in fig 1 are bordering the forest and agricultural land. The TSS results of A and B do not vary substantively as those of point E which is the main outlet of the study site. Since most of the intensive
agriculture is happening below points A and B towards point E, this study used the TSS results of point E to interpret the effectiveness of PES approach in delivering the desired ecosystem services. Therefore the TSS results at point E are both contribution of the forested lands (which can be treated as '*nature contribution*') and the agricultural land. From Fig 6, it is observable that the PES approach helped in reducing the erosion levels to almost what the nature is currently contributing. It is also paramount to note that there are undocumented few cases where
farmers are doing what we can call '*illegal farming*' inside the forest. This was not accounted for in this study.

The PES-farmers are incentivized by a compensation of 30% of the total cost of implementing the recommended SLM technologies in their farms. The implementation is on-going with each farmer implementing his/her LMP. So far, the compensation is estimated at an average of KES 4,541 (appx 45 US$) per household.

35  The calibrated SWAT model in this study provides a system that allows assessment of effectiveness of PES approach in the delivery of desired ecosystem services downstream. At the onset of the PES project, the average TSS was estimated at 71.05mg/L. This means the PES project did not start from scratch (from the worst scenario) where average TSS is estimated at 124.15 mg/L by the SWAT model. Since there existed some SLM technologies through farmers own initiatives, this value of TSS before the onset of PES project is assumed to be the value under the conventional approaches. PES approach is a unique scheme as it accelerates adoption of SLM technologies
40  through incentives. This was proved in this study, as the PES pilot project led to an increase of SLM adoption from an estimated 11% to 32% in the study site (table 4). This translated into an improvement in water quality as average TSS reduced from 71.05mg/L to 42.73mg/L. The SWAT model predicts a worst scenario of an average TSS of 124.15 mg/L. The SWAT model also predicts the best scenario at an average TSS of 12.76 mg/L. Figure 4 shows the different scenarios starting with the predicted worst scenario to the best (target) scenario.

45  This study utilized TSS as the key proxy indicator in assessing the effectiveness of PES approach within the first year. Other indicators for instance stream recharge in dry periods, land productivity, infiltration rates etc take longer periods before they manifest. This requires longer study periods which could affect the results observed in this study. It is important to note that SWAT model was calibrated and validated using data observed (for 71 days) in March – May 2016. This denotes lack of long term data for the study site to calibrate over a longer period which
50  may affect the consistency of final results.





SLM practices were represented by modifying model parameters to predict the best and worst scenarios. Fitting of actual measured values in between SWAT predicted scenarios is subjective and may vary with investigation methods of the measured values. In this study however, watershed management through PES approach is seen to accelerate adoption of SLM technologies which led to a reduction in TSS downstream.

5 The method presented in this study should also be validated on other sub-watersheds of different scales within the Sasumua watershed. The results presented in this study are from one sub-watershed of approximately 6.08km$^2$ out of possible 28 sub-watersheds. Thus, this study would be strengthened by validating the effectiveness of PES approach at multiple discretization levels and spatial scales within the Sasumua watershed.

## 4    Conclusions

10 PES pilot project did not start from scratch as the average TSS observed at the onset of the study was 71.05mg/L compared to predicated worst scenario of an average of 124.15mg/L. This is expected as some farmers adopt SLM technologies through conventional approaches. This study established adoption of SLM technologies through conventional approach at 11%. However, remarkable changes were observed after one year of KAPLSMP PES project implementation where the adoption rate improved from 11% to 32% within the first year. This drastically 15 reduced the observed average TSS from 71.05mg/L to an average TSS of 42.73mg/L. Watershed management through PES scheme is therefore identified as an effective approach in accelerating sustainable land management practices. However, long term research data is highly recommended to validate the effectiveness of PES over number of years especially on ecosystem services that manifest after long periods and establishing whether PES incentives actually maintain best conditions at farm level. More ecosystem services should also be monitored to 20 validate the TSS results.

**Acknowledgements**

The authors thank Dr. Karim Abbaspour for his commendable assistance with SWAT especially in running the new semi-automated calibration engine (SWAT-CUP). We also recognize Word bank through KAPLSMP for funding this research project.




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





**Table 1: Chronology of KAPSLMP PES project implementation in Sasumua watershed**

| Date | KAPSLMP activities | M.sc research project activities |
|---|---|---|
| | *Exploratory studies* | |
| 2005/2006 | PES project conceptualization | |
| 2008/2009 | PES feasibility study by ICRAF/PRESA | |
| 2009 | Situational analysis study | |
| 2011 | A study to evaluate the impacts of soil and water conservation practices on ecosystem services in Sasumua watershed, using SWAT model | |
| 2013 | The World Agroforestry Centre (ICRAF) in partnership with Jomo Kenyatta University of Agriculture and Technology (JKUAT) sought to explore the potential of using Payment for Ecosystems Services (PES) through its program, Pro-poor Rewards for Environmental Services in Africa (PRESA) | |
| 2014 | Policy and institutional analysis for PES | |
| 2014 | A study to assess water quality status of Sasumua Watershed (Baseline Results of key monitoring points) | |
| 2014 | A study to evaluate the potential for Payment for Ecosystem Services (PES) and policy implications in Sasumua watershed. | |
| *Actualizing 'theory into practice'* | | *M.sc research project activities* |
| February - March 2015 | Preliminary meetings with key stakeholders | M.sc Research Proposal and establishment of a representative watershed |
| March - May 2015 | Awareness creation on PES pilot project to potential small scale farmers, formation of CIGs and training | Baseline on water quality in the study site |
| June 2015 | Setting out - Establishment of individual Land Management Plans (LMPs) including field demonstrations | Baseline on SLM status in the study site |
| October 2015 | Training PES sub-committee on data collection | |
| October 2015 | Signing of PES contracts between the ES buyers and the sellers | |
| October - December 2015 | | Rainy season (Water quality sampling) |
| March 2016 | Auditing for PES rewards | |
| April 2016 | Rewarding of the leading farmers | |
| March - May 2016 | | Rainy season (Water quality sampling) |
| June 2016 | | Field study to establish SLM progress in the study site after one year of PES implementation |

**Table 2: Land use in the representative sub-watershed**

| Land use | Area |
|---|---|
| Forest | 87.6 % |
| Pasture | 0.7 % |
| Agriculture | 11.7 % |





**Table 3: Model input data information**

| Data type | Source | Description |
|---|---|---|
| DEM | USGS website | 30 m resolution, U.S. Geological Survey (USGS) |
| Soils | SOTER | 1:1,000,000 Soil data was extracted from the Digital Soil and Terrain Database of East Africa (SOTER). |
| Land use | USGS website | Landcover maps were generated from 2016 landsat image obtained from USGS website using maximum likelihood method classification |
| Weather (1970-2014)[a] | Weather records for three stations in the watershed; Agricultural training centre (9036152), South Kinangop forest station (9036164) and Sasumua dam station (9036188) | Minimum and maximum daily temperature, daily precipitation, relative humidity, solar and wind |
| Rainfall[b] | Observed between January and May 2016 | Daily precipitation |
| Crop management | Field survey, (Mwangi et al., 2014) | Interviews with key informants and individual farmers on historical and current field crops and reviewing available literature. |
| Stream flow | Observed during March 18th – May 27th | Daily stream flow (m3/s) |
| Water quality | Observed during three rainy seasons March – May 2015 October – December 2015 March – May 2016 | Daily Total Suspended Solids (TSS) |

[a]Weather data between 1970 and 2014 was used in generating a weather generator data (WGN) for the study site using WGNmaker4 (Boisrame, 2016). This was particularly important in simulating the weather data that was not actually measured during the study period e.g solar, wind, relative humidity etc.

5     [b]The rainfall observed during the study period was used in SWAT calibration and in focusing the modelling to specific period of simulation (starting and ending dates) during which actual data collection on flow and Total suspended solids (TSS) was carried out.

**Table 4: Progress of SLM adoption through the PES pilot project**

| | A | B | C | D | E | F |
|---|---|---|---|---|---|---|
| SLM Technology | Baseline status | After 1 year | Baseline status $= A/F$ | After 1 year $= B/F$ | Change $= \dfrac{B-A}{F}$ | Target |
| Terrace (M) | 820 | 1,561 | 9% | 18% | 8% | 8,820 |
| Retention ditch (M) | 120 | 551 | 4% | 18% | 14% | 3,120 |
| Grass strip (M) | 920 | 3725 | 10% | 42% | 32% | 8,820 |
| Riverbank protection (M) | 210 | 520 | 21% | 51% | 31% | 1,015 |
| No. of Napier splits | -- | 10865 | -- | 47% | -- | 22,925 |
| No. of Forest trees | -- | 183 | -- | 15% | -- | 1185 |
| No. of Fruit trees | -- | 1095 | -- | 96% | -- | 1145 |
| Estimated SLM adoption | | | 11%[a] | 32%[a] | 21% | |

[a]Only Terraces, Retention ditches, Grass strips and Riverbank protection were considered (proxy indicators of SLM

10     adoption) as they were easily measurable in baseline and the current status.



**Table 5: Model parameters considered in SWAT-CUP calibration**

| Parameter | Description | Minimum-maximum | Default values | Final calibrated values |
|---|---|---|---|---|
| *Parameters sensitive to Discharge* | | | | |
| CN2.mgt | Initial SCS runoff curve number for moisture condition II | 35 - 98 | 83 | 81.4 |
| ALPHA_BF.gw | Base flow alpha factor (1/days) | 0 - 1 | 0.048 | 0.6 |
| GW_DELAY.gw | Ground water delay time (days) | 0 - 500 | 31 | 19.9 |
| GWQMN.gw | Threshold depth of water in the shallow aquifer required for return flow to occur (mm $H_2O$) | 0 - 5000 | 1000 | 2809.3 |
| ESCO.hru | Soil evaporation compensation factor | 0 - 1 | 0.95 | 0.1 |
| EPCO.hru | Plant uptake compensation factor | 0 - 1 | 1 | 0.6 |
| SOL_K.sol | Saturated hydraulic conductivity (mm/hr) | 0 - 2000 | 65 | 252.7 |
| SOL_AWC.sol | Available water capacity of the soil layer (mm $H_2O$/mm soil) | 0 - 1 | 0.3 | 0.2 |
| SLSUBBSN.hru | Average slope length (m) | 10 - 150 | 60.98 | 74.8 |
| CH_K2.rte | Effective hydraulic conductivity in main channel alluvium (mm/hr) | -0.01 - 500 | 0 | 27.9 |
| OV_N.hru | Manning's N value overland flow | 0.01 - 30 | 0.14 | 29.9 |
| HRU_SLP.hru | Average slope steepness (m/m) | 0 – 0.6 | 0.0766 | 0.5 |
| *Parameters sensitive to sediments* | | | | |
| SPEXP.bsn | Channel re-entrained exponent parameter | 1 - 1.5 | 1 | 1.2 |
| SPCON.bsn | Channel re-entrained linear parameter | 0.0001 - 0.01 | 0.0001 | 0.00237 |
| CH_EROD.rte | Channel erodability factor | 0 - 1 | 0 | 0.2 |
| CH_COV.rte | Channel cover factor | -0.001 - 1 | 0 | 0.6 |
| USLE_P.mgt | Support practice factor | 0 - 1 | 1 | 0.8 |





**Table 6: Selected land management operations with their adjusted parameters to reflect best and worst scenarios in SWAT**

| Main SLM Technologies promoted under PES pilot project | Land Management selected for representation in SWAT | Adjusted Parameters in SWAT | Description | Parameter ranges | Adjusted to reflect Best scenario | Adjusted to reflect Worst scenario |
|---|---|---|---|---|---|---|
| Fanya Juu terraces and retention ditches | Terraces | TERR-P | To reflect reduced sediment losses | 0-1 | 0.18 | 1 |
| | | TERR-SL | To represent the minimum distance between the terraces in meters | 0-100 | 5 | 100 |
| | | TERR-CN | To account for increased infiltration | 20-100 | 76 | 99 |
| Contour farming | Contour planting | CONT_CN | To account for increased surface storage and infiltration | 20-100 | 76 | 99 |
| | | CONT_P | To account for decreased erosion | 0-1 | 0.18 | 1 |
| Grass strips | Filter strip | VFSRATIO | Ratio of field area to filter strip area | 0-300 | 30 | 300 |
| | | VFSCON | Fraction of the HRU which drains to the most concentrated 10% of filter strip | 0.25-0.75 | 0.75 | 0.25 |
| | | VFSCH | Fraction of the flow within the most concentrated 10% of the filter strip which is fully channelized | 0-100 | 5 | 100 |
| | | FILTERW | To account for increasing trapping efficiency of the filter strip | 0-100 | 5 | 0 |
| Planting of fruit/fodder trees along the contour (deep rooted) and planting crops (shallow rooted) in between the strips | Strip cropping | STRIP_N | To represent increased surface roughness in the direction of flow | 0.001-0.5 | 0.5 | 0.001 |
| | | STRIP_CN | To account for increased infiltration | 20-100 | 76 | 99 |
| | | STRIP_C | To reflect the average value of the multiple crops within the field | 0-1 | 0.4 | 0.4 |
| | | STRIP_P | To account for decreased erosion | 0-1 | 0.18 | 1 |





**Table 7: Land cover percentages in the study site**

|  | *Sub-basin 1* | *Sub-basin 2* | *Sub-basin 3* |
|---|---|---|---|
| *Forest* | *351.52 Ha (96.7%)* | *174.95 Ha (86.4%)* | *6.03 Ha (14.4%)* |
| *Pasture* | *0.58 Ha (0.2%)* | *1.22 Ha (0.6%)* | *2.28 Ha (5.4%)* |
| *Agriculture* | *11.50 Ha (3.1%)* | *26.25 Ha (13%)* | *33.63 Ha (80.2%)* |
| *Total* | *363.60 ha* | *202.42 ha* | *41.94 ha* |

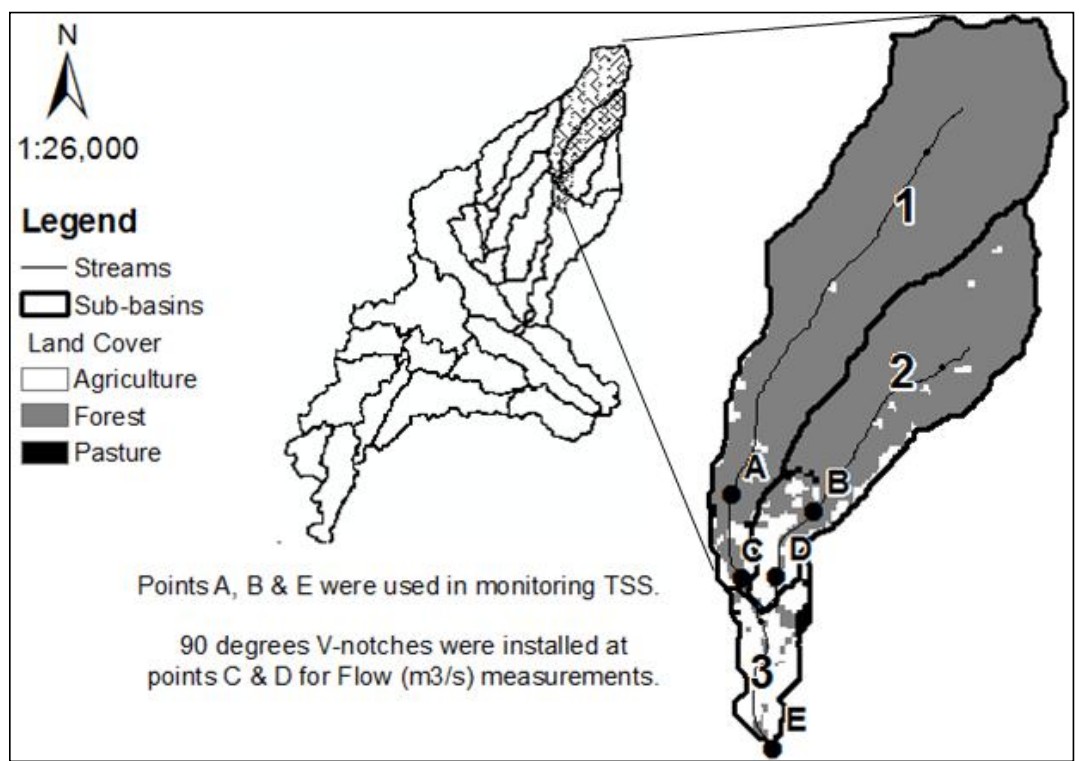

**Figure 1: The representative sub-watershed (headwater)**





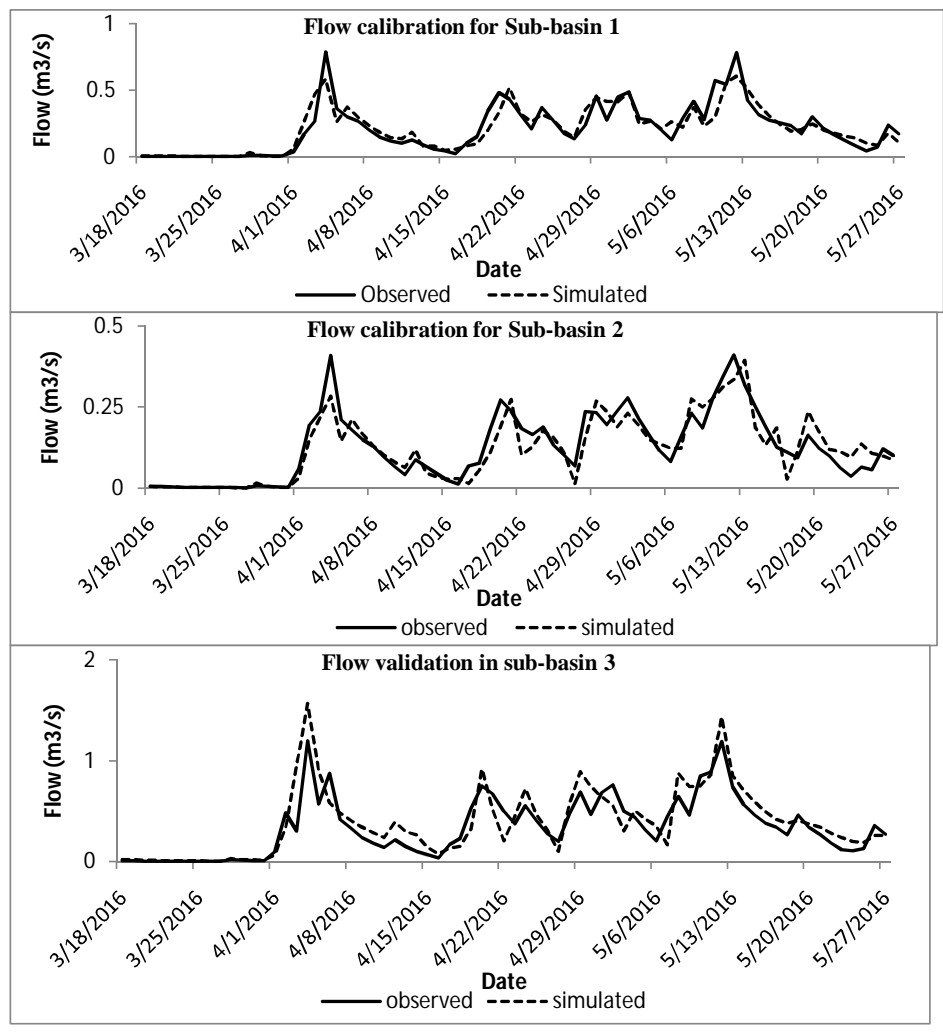

**Figure 2: Flow calibration and validation**





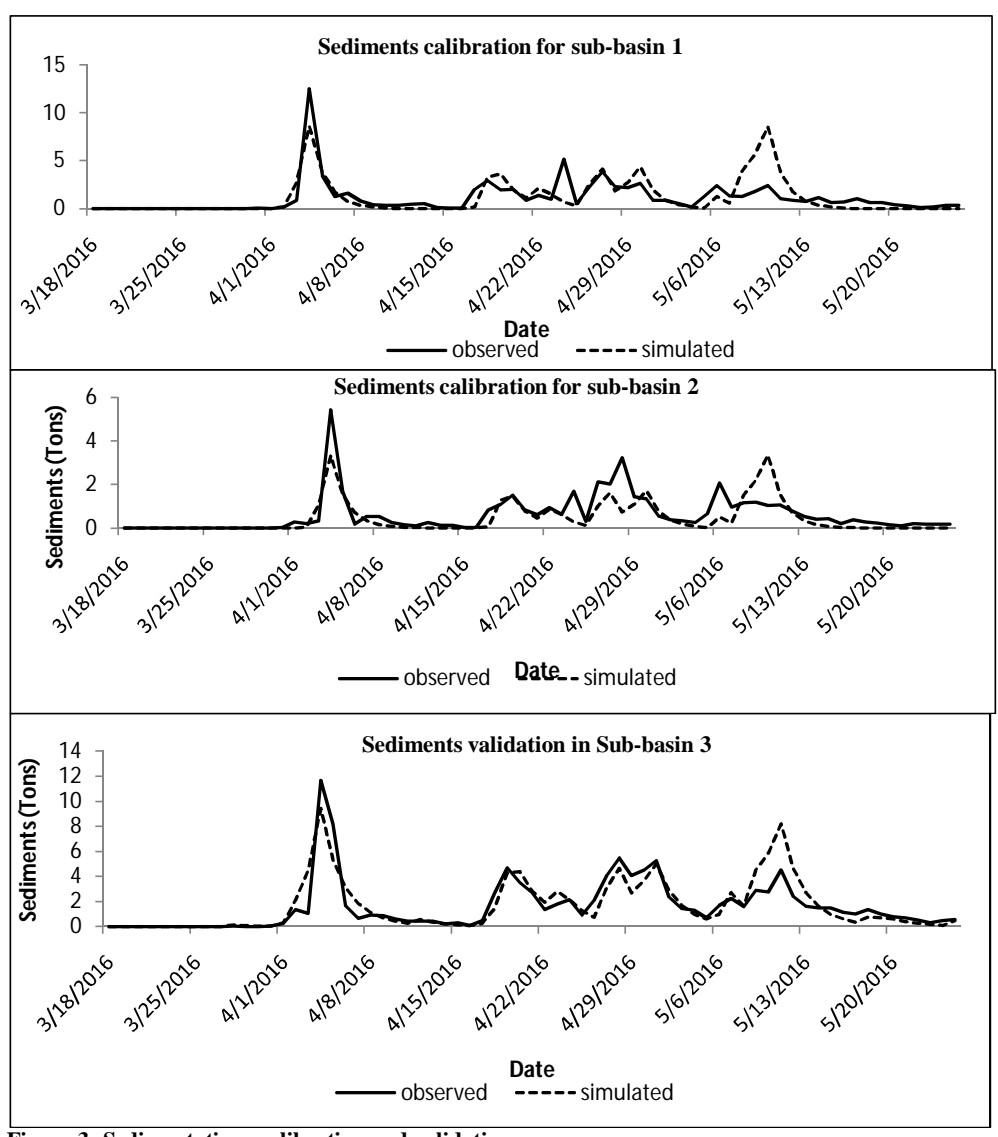

**Figure 3: Sedimentations calibration and validation**




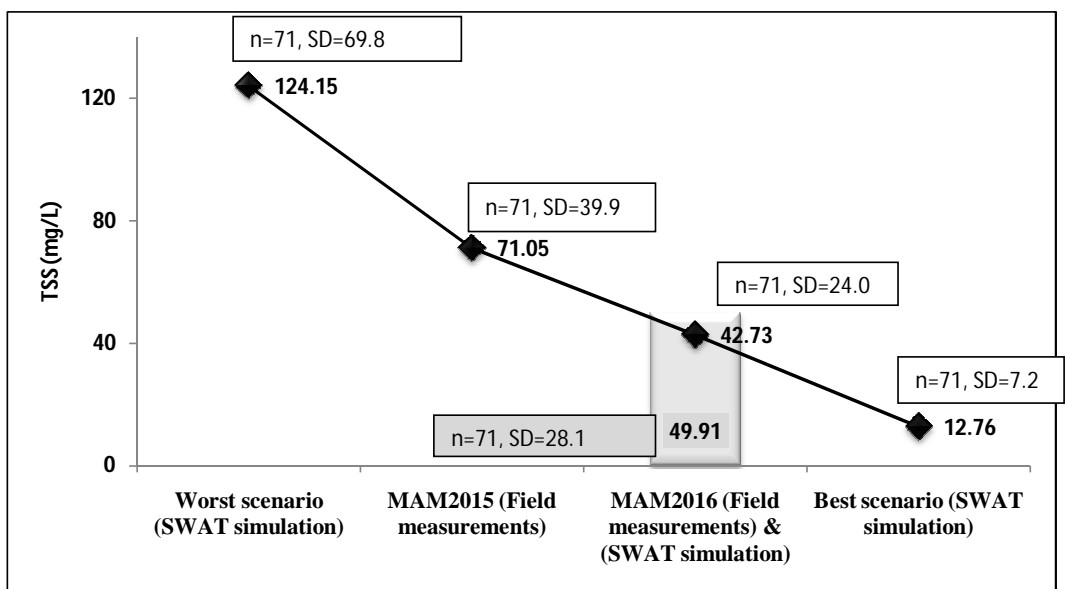

**Figure 4: Average Total Suspended Solids (TSS) in rainy seasons**

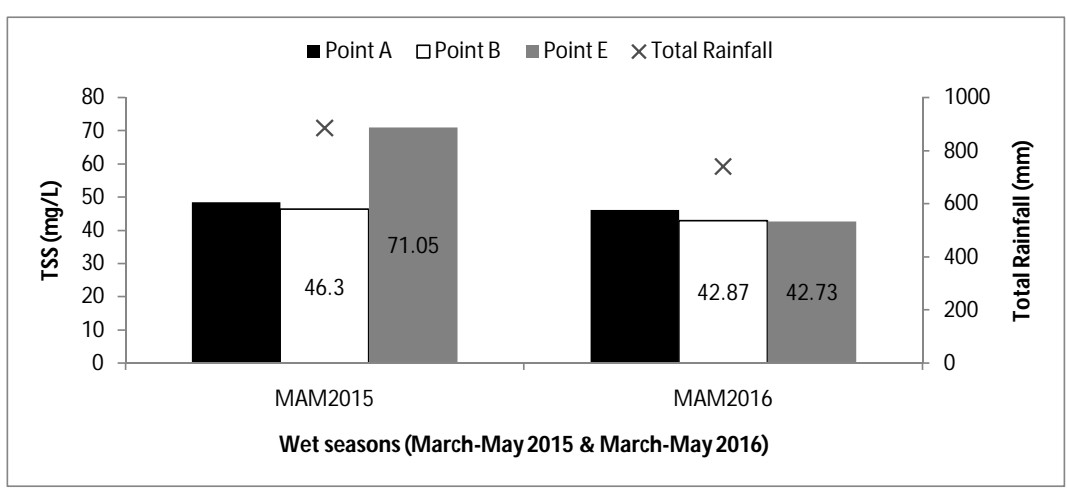

**Figure 5: TSS results of the three points (A, B & E) during the two rainy seasons of March-May 2015 and March-May 2016**





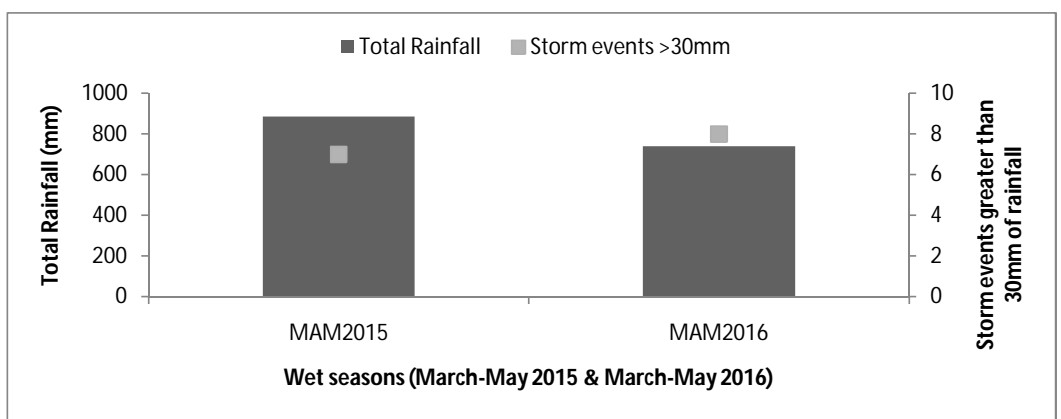

**Figure 6: Rainfall series for both March-May 2015 and March-May 2016**

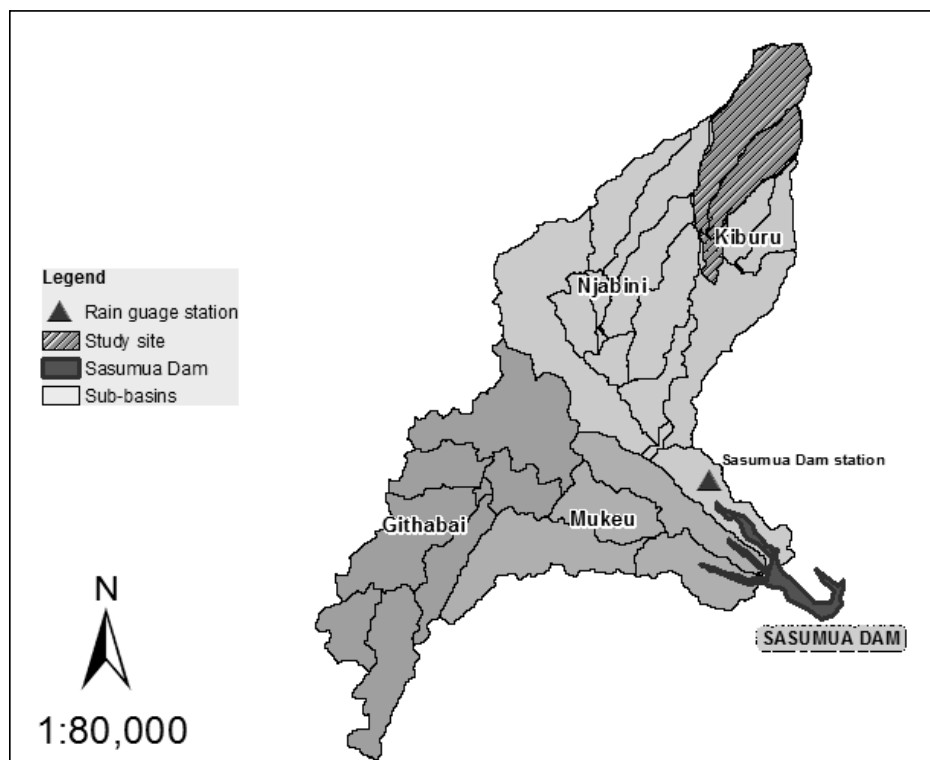

**Figure 7: Sasumua Dam Rain gauge station**





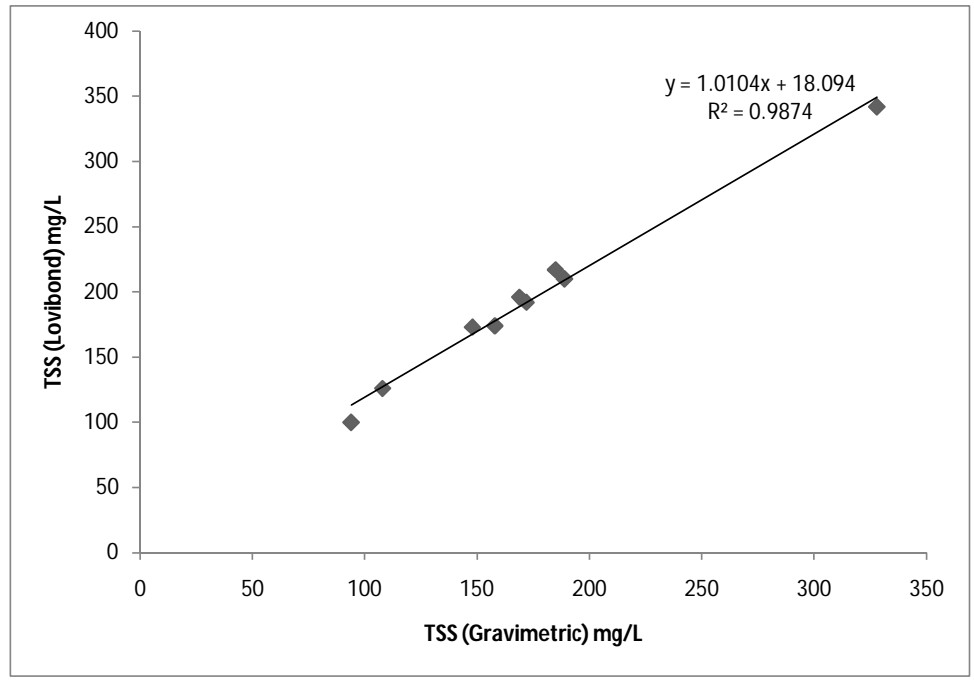

**Figure 8: The relationship between Lovibond TSS and gravimetric TSS**