# Peer review of "Assessment of the effectiveness of Payment for Ecosystem Services (PES) in the delivery of desired Ecosystem Services in Sasumua catchment, Kenya"

_Hydrology and Earth System Sciences, 2016_

## Referee Comment (RC1) · Anonymous Referee #1 · 5 Dec 2016

The article describes a very recent study on PES in a small Central Kenyan watershed. Total suspended solids (TSS) in the river are monitored during two rainy seasons 2015 and 2016, one before and one after implementation of sustainable land management (SLM) practices, and simulated by SWAT.

The description of materials and methods focuses on field data collection and parameterization of SWAT to represent the SLM. Because observing approximately 40% less TSS in the March-May rainy season in 2016 as compared to 2015, it is concluded that the PES scheme is effective in accelerating SLM practices.

[Figure]

While the title suggests a focus on the PES concept, the article is essentially only about monitoring and simulating the impact of some SLM practices on the reduction of TSS in river runoff, i.e., land management practices to reduce soil erosion. And even this is only poorly founded on the comparison of only two rainy seasons – one before and one after implementation of SLM. In the whole article, I cannot see any attempt of an "ecosystem" approach, meaning "a community of living organisms in conjunction with the nonliving components of their environment".

Instead of advising on details of the article, I recommend to reconsider the main purpose of the article: If the title remains the same, I would expect a thorough discussion of the policy issues (magnitude of the incentives, conditions for getting payments, technical and/or scientific support of implementation, training of farmers, long term maintenance, etc.), and less on the technical details of monitoring and simulating the physical processes.

Should the focus turn to monitoring and modelling the physical effects of land management practices – which is still worth publishing – then, in my opinion, more data are required. Most land management practices do not have their final effect right in the first season after implementation, but need some time to stabilize.

Recommendation: reject the article, because title and contents do not match. Consider to re-submit either a policy focused article according to the current title, or focus on the monitoring and simulation issues with a title like "Monitoring and modelling the effect of sustainable land management practices on soil erosion". Or both.

---

## Referee Comment (RC2) · J. R. Kosgei (Referee) · 23 Dec 2016

[revised manuscript text omitted]

---

## Author Comment (AC1) · 4 Jan 2017

[revised manuscript text omitted]

This study was established under the pilot project to assess the effectiveness of PES scheme in the delivery of desired ecosystem services in Sasumua catchment. The paper describes the PES concept implemented in the Sasumua watershed. A representative sub-watershed was also selected to model and predict the effect of sustainable land management practices on soil erosion – a reflection of what is happening under the PES scheme.

**2 Materials and Methods**

**2.1 Site description**

Sasumua is located in Central Kenya, southern ridges of Kenya's Aberdare Mountains about 90 km northwest of Nairobi, at an altitude of 2200-3850 m.a.s.l. Mean annual rainfall is 1000-1600 mm, peaking March-May and October-December in a binomial pattern (Gathenya *et al*, 2010). The catchment of the Sasumua reservoir is 107 km$^2$ and comprises three sub-catchments: Sasumua (67.44 km$^2$) Chania (20.23 km$^2$) and Kiburu (19.30 km$^2$).

A representative sub-watershed was strategically selected in a way possible to correlate the SLM adoption to water quality improvement – headwater sub-watershed. It was difficult to monitor all PES farmers in the larger Sasumua watershed given time and the resources available. With a small sub-watershed, it was practicable to follow the progress of every farmer. The headwater sub-watershed is approximately 6.08km$^2$ with a total of 41 farmers engaged in PES pilot project. The selection was strategic such that all farmers had the same outlet point for monitoring reduction in sediments loads as they adopt new SLM practices (figure 1). Furthermore, the choice of a headwater ensured there were no interferences on the downstream results by upstream activities allowing a direct relationship of the water quality status to the actual progress made on adoption of new SLM technologies.

**2.2. KAPSLMP PES pilot project**

KAPSLMP PES pilot project is a public scheme where the government represented by Kenya Agricultural Productivity and Sustainable Land Management Project (KAPSLM) is buying Ecosystem services as a dummy ES buyer. Table 1 shows chronology of KAPSLMP PES project implementation in Sasumua watershed. In June 2015, Kenya Agricultural and productivity and Sustainable Land Management Project (KAPSLMP) initiated a PES pilot project in Sasumua. This was to actualize '*theory into practice*' from past study findings and recommendations.

It was estimated that Sasumua watershed yielded 40,934 tons of sediments per year most of which are deposited at the intakes where they cause siltation problems and in Sasumua reservoir where they increase water turbidity and thus high water treatment costs (Mwangi, 2014). In 2014, the total cost for water treatment in Sasumua was estimated at KES 30,592,640 (approximately USD 305,926) annually. This translates to a cost of KES 747 (approximately USD 7.5) per ton of sediments generated. The cost of treating water at Sasumua treatment plant was estimated at Ksh. 2,080 (USD 208) per 1,000 m$^3$ against Ksh.1,120 (USD 112) per 1,000m$^3$ at Ndakaini dam (Mwangi, 2014). This is attributed to poor raw water quality at Sasumua reservoir among other factors.

The annual costs of water purification were predicted to go up by Ksh 15,534,720 (approximately USD 155,347) by 2024 if farming practices continue as they are (PRESA, 2014). This problem can be solved by developing partnerships with land owners to implement sustainable land management practices such as terraces, contour farming, grass filter strips and grass waterways through an alternative approach. In this case PES was recommended as an alternative approach in Sasumua watershed (PRESA, 2014).

**2.3. The goal of PES pilot project**

The goal of KAPSLM PES pilot project was to demonstrate effectiveness of PES as tool in delivering desired ecosystem services in Sasumua catchment. The desired ecosystem services are improved water quality downstream as one of the provisional services of a reconstructed ecosystem. The immediate objectives were to promote pre-determined SLM technologies that would significantly reduce soil erosion in the catchment – consequently reducing the sediments downstream. The pilot project targeted hotspot areas identified in past studies (Mwangi 2014). Table 2 describes the targeted SLM technologies to be piloted by approximately 1,017 farmers in Sasumua watershed.

**2.4. Estimated benefits from PES project in the watershed**

The past studies in the Sasumua watershed found that maximum benefits in terms of soil erosion reduction will be realized only if conservation measures are targeted in the pre-identified hotspot areas. According to (Mwangi, 2014) the following were some of the expected benefits under a PES scheme:

- Implementing a grassed waterway approximately 20 kilometres long and 3 metres wide will reduce soil sedimentation by 20%.
- A combination of terraces and grassed waterways would reduce sediment inflow by 75%.
- 10 metres filter strips and grassed waterway will reduce sedimentation by 73%.
- Savings in water treatment and costs under PES were estimated to be KES 20,497,040 (approximately USD 204,970).

**2.5. The ES buyer and sellers**

KAPSLMP was acting as a dummy buyer of Ecosystem Services (ES) at the initial pilot stage. Prove of PES viability was required to enable replication and scaling up of the PES project. Therefore, assessing effectiveness of PES as an alternative approach to soil erosion control and water quality was an integral part of the KAPSLMP PES project.

The ES sellers are small scale farmers located in the hotspot areas. The PES farmers are organized into Common Interest Groups (CIGs). The CIGs include various agricultural value chains being promoted under KAPSLM project including; Tree tomato, Dairy, Irish potatoes, straw berry, Bee keeping, Agroforestry, Fish farming etc. Farmers receive trainings through the respective CIGs. At the time of this study, the KAPSLMP PES project had been implemented in an area covering approximately 762.4 hectares with a total of 1,017 PES farmers.

**2.6. Conditions for getting payments**

Land Management Plans (LMPs) were established for all PES farmers in the watershed. The SLM experts and the extension agents agreed with each farmer what measures would be implemented and supported in each farm to reduce soil erosion. Individual farmers were given a copy of an LMP in form of a map – describing areas within the farm where each type of recommended SLM technology will be implemented.

To qualify for PES payments, the farmers had to implement the SLM technologies as per specifications in the LMPs. They were also to maintain the LMPs regularly. This entailed taking care grass strips, trees, keeping the terraces stable etc. The PES program compensated 30% of the total SLMP implementation cost incurred by the farmer.

**2.7. Training of farmers**

Through the Common Interests Groups (CIGs), PES farmers received weekly training based on topics relevant to the agricultural value chains they were involved in. The aim was to promote good husbandry practices and increase production per unit area while conserving the environment. The training sessions were led by the extension service providers contracted under the KAPSLM project. The service providers were also available for consultation by the farmers. They organized on-farm demonstrations and audited the progress of SLM adoption in order to assess the compensation for each farmer.

**2.8. Long term maintenance**

In each of the 4 micro-catchments, the PES participants elected PES committee members. The PES committee worked closely with Water Resources Users Association (WRUA). WRUA are associations of water users and riparian land owners who have associated for the purposes of cooperatively sharing, managing and conserving a common water resource. WRUAs are part of catchment management strategy established under Water Act (2002) of Kenya. The PES committee members are the leaders of the PES scheme in their respective micro-catchments. They were trained on PES concept, good husbandry practices, environmental conservation etc. They were supposed to lead as examples to other farmers in all PES related activities promoted under the KAPSLM PES project. The PES committee members conducted regular on-farm checks in their respective micro-catchments to encourage adoption of SLM technologies. They represented farmers through forwarding their needs and priorities during PES stakeholder meetings. They provide a communication link between the KAPSLM PES management team and the farmers. The involvement of PES committee together with the Water Resources Users Association (WRUA) in the PES project activities was expected to maintain the farms in their desired status during and beyond the KAPSLM PES pilot project.

**2.9. Monitoring and modelling the effect of sustainable land management practices on soil erosion**

A small area was established as a representative sub-watershed to study what is happening under the PES scheme. This was to make it possible to follow the individual actions of all farmers in the study site for deeper analysis on the effect of additional SLM technologies to downstream water quality. The study site is a headwater – stretches from the forest towards cultivated lands. The headwater comprises of 3 sub-basins covering an area of 6.08 km$^2$ whereby 42.3 ha are under intensive cultivation. As shown in Fig 1, Forest is the dominant land cover for sub-basins 1 and 2 while agriculture is the dominant land cover in sub-basin 3. Table 3 shows characterization of the study site. There are a total of 67 farmers of which 41 are participating in the PES pilot project. The key point to note is that sub-division of land is high in this area, increasing the number of individuals who own land (some of whom are

'absent farmers[1]'). This number (67) is estimated from the farms with title deeds and not individuals. The actual number of individuals could be higher including the 'absent farmers[1]'.

The article looks at PES scheme as an alternative tool to conventional approaches in regard to SLM adoption. PES scheme incentivizes farmers to adopt and continue with certain practices away from the *'norm'*, and as a result influence the water quality status downstream. The article recognizes that most of the ecosystem services take time to manifest. Total Suspended Solids (TSS) was selected as a proxy indicator as any immediate change on erosion (e.g terraces/retention ditches trapping sediments) will be detected if reference data is available (baseline data).

The conditions (SLM status and water quality status) before PES scheme are assumed as conditions under conventional approach. (*Assumption: Farmers would still be under these conditions today if no alternative was provided*). Data on status of SLM adoption and TSS were determined before onset of PES pilot project to represent status of water quality under *'norm'* conditions. The conditions after one year of PES pilot project are assumed as improved conditions achieved under PES scheme – as an alternative approach. It should be noted that PES scheme is a continuous project and the results presented are those monitored in the first year.

Since long term data was not available over a considerable period (e.g hydrology data of over 20 years), this study adopted SWAT model to establish a model-system of the study site. This was done through developing best and worst scenarios. This allowed fitting field measurements in between best and worst scenarios to help determine the inclination of the observed scenarios – whether positive towards the target scenario or negative towards the worst scenario. The modelling in this study helps to predict what the end would look like.

**2.9.1. Description of SWAT Model Inputs**

[revised manuscript text omitted]

**3. Results and discussions**
**3.1. Overall progress of SLM adoption**
This study sought to establish the progress made by the KAPSLMP PES project in all the four micro-catchments (Kiburu, Njabini, Githabai and Mukeu). The average progress was established to be 45.5% towards the targeted status (Table 2) (i.e. cumulative progress vs. the target). In general, grass strips and terraces achieved the highest achievement with 59.9% and 56.5% respectively (figure 5).

**3.2. Magnitude of the incentives**
The PES-farmers are incentivized by a compensation of 30% of the total cost of implementing the recommended SLM technologies in their farms. The implementation is on-going with each farmer implementing his/her LMP. So far, the compensation is estimated at an average of KES 4,541 (approximately USD 45) per household. Approximately 41% of PES farmers have been evaluated and found to have implemented the SLM technologies as stipulated in the LMPs. The total estimated costs of SLM implementation by the 41% of PES farmers are estimated at KES 6,236,887 (approximately USD 62,369). At the time of this study, the PES farmers had received a compensation (30% of total costs) worth of KES 1,871,066 (approximately USD 18,711).

**3.3. Simulated impact of PES scheme on water quality**
TSS was used as the main proxy indicator in assessing the effectiveness of PES approach. The comparison of photometric method and the conventional gravimetric method in TSS measurements showed that the relationship is linear (Fig 6). Generally, the lovibond reads a relatively higher TSS value from that of gravimetric method with an average factor of 1.12. This means if conventional gravimetric method reads (X), the Lovibond reading can be estimated at (1.12X).

At the onset of PES pilot project, the SLM adoption status was established at 11% – these included; 820m of terraces, 120m of retention ditches, 920m of grass strips, and 210m of river bank protection (Table 7). The water quality status before the onset of the PES pilot was represented by TSS at an average of 71.05 mg/L. After one year of PES project implementation, the SLM adoption status improved to 32%; 1561m of terraces, 551m of retention ditches, 3725m of grass strips, and 510m of river bank protection. At this period, the average TSS observed was an average of 42.73 mg/L. T-test to compare seasonal rainfall of March – May 2015 and March – May 2016 showed that there were no significant differences on daily rainfall characteristics of the two seasons (Table 9). However, the reduction of TSS from 71.05 mg/L to 42.73 mg/L was identified as significant (Table 10); a result which would have not been realised if no alternative was provided.

Figure 7 shows the TSS results in the study site measured during two rain season of March-May 2015 and March-May 2016. As aforementioned, the study site is a 'headwater' (fig 1) – points A and B in fig 1 are bordering the forest and agricultural land. Since most of the intensive agriculture is happening below points A and B towards point E, this study used the TSS results of point E to interpret the effectiveness of PES scheme as an alternative approach in delivering the desired ecosystem services. The TSS results at point E are both contribution of the forested lands (which can be treated as '*nature contribution*') and the agricultural land. From Fig 7, it is observable that TSS reduced to almost what the nature is currently contributing. This is attributed to the alternative approach provided – the PES as tool in watershed management. The PES incentives are expected to further improve and maintain the best conditions at farm level which will assure sustained delivery of the desired ecosystem services. It is paramount to note that there are undocumented few cases where farmers are practising '*illegal farming*' inside the forest. This was not accounted for in this study.

The calibrated SWAT model in this study provides a model-system that allows assessment of effectiveness of PES approach in the delivery of desired ecosystem services downstream. At the onset of the PES project, the field observation of TSS was estimated at an average of 71.05mg/L. This means the PES project did not start from the worst scenario which is predicted at 124.15 mg/L by the SWAT model. Since there existed some SLM technologies through farmers own initiatives, this value of TSS observed before the onset of PES project is treated as the value under the conventional approaches – at '*norm*' situations. The initial findings after one year demonstrate that an improvement on water quality had actually occurred – average TSS significantly reduced from 71.05mg/L to 42.73mg/L. The SWAT model predicts that the TSS would reduce to an average TSS of 12.76 mg/L if the representative sub-watershed is managed to its best scenario – conditions to be realized if all farmers implement the land management plans as stipulated. Figure 2 shows the simulated impact of PES scheme to water quality in the study site.

**4. Conclusions**

The conditions observed after one year suggest improved conditions achieved under an alternative approach. The results observed in the representative sub-watershed imply an improvement in the adoption of SLM technologies. The increased adoption of SLM technologies are attributed to the PES scheme as an alternative approach. It may not be sufficient to compare and conclude on water quality improvement based on two field measurements observed before and after one year, but fitting the field observations in the SWAT model-system illustrates a positive trend towards delivery of the desired ecosystem services. It should be noted that the results presented in this article are not ultimate results rather findings monitored after one year to represent improved conditions (out of *'norm'*) achieved under PES scheme as an alternative approach to watershed management. If the initial findings are something to go by, then, it can be said that PES scheme played a fundamental role in improving water quality status from approximately 48% to 73% towards the targeted status according to the SWAT simulation.

From the field observations, it is evidently clear that best conditions of a watershed cannot be realised if farmers are left to continue under the *'norm'* conditions. Alternative approaches to watershed management are inevitable. This study demonstrates that increased adoption of SLM technologies are possible under PES scheme which consequently can lead to provision of desired ecosystem services. The SWAT model predicts that the TSS would improve to desired status if the representative sub-watershed is managed to its best scenario. These conditions can only be realized if all farmers implement SLM technologies as recommended. PES incentives are expected to encourage full implementation of the land management plans.

The results in this study suggest that PES pilot project did not start from scratch as the average TSS observed at the onset of the study was 71.05mg/L against worst predicated scenario of an average of 124.15mg/L. This is an indication that some farmers adopt SLM technologies through their own initiatives – possibly through voluntarily attending to trainings, education background, through extension workers, media adverts etc. This way, there are some farmers who initiate own conservation measures without necessarily being incentivised. In this study, the own initiatives are regarded as conventional approaches. It was established that adoption of SLM technologies through conventional approach stood at 11% before the PES scheme.  However, remarkable changes were observed after one year of PES project implementation where the SLM adoption improved to 32% within the first year. Watershed management through PES scheme is an alternative approach that should be promoted widely. PES scheme is seen as an effective approach in accelerating sustainable land management practices which can lead to improved conditions downstream.

**Areas of further research**

Long term research data is highly recommended to validate the effectiveness of PES over number of years especially on ecosystem services that manifest after long periods and establishing whether PES incentives actually maintain best conditions at farm level. More ecosystem services should also be monitored to validate the TSS results. The study utilized TSS as the key proxy indicator in assessing the effectiveness of PES approach within the first year. Other indicators for instance stream recharge in dry periods, land productivity, infiltration rates etc take longer periods before they manifest. This requires longer study periods which could affect the results observed in this study.  SWAT model was calibrated and validated using data observed (for 71 days). This denotes lack of long term data for the study site to calibrate over a longer period which may affect the consistency of final results. SLM practices were represented by modifying model parameters to predict the best and worst scenarios. Fitting of actual measured values in between SWAT predicted scenarios is subjective and may vary with investigation methods of the measured values. However, this study demonstrates a positive trend towards the target scenario. The method presented in this study should also be validated on other sub-watersheds (including non-headwater) of different scales within the Sasumua watershed. The simulation results presented in this study are from one sub-watershed of out of possible 28 sub-watersheds. Thus, this study would be strengthened by validating the effectiveness of PES approach at multiple discretization levels and spatial scales within the Sasumua watershed.

**Acknowledgements**
The authors thank Dr. Karim Abbaspour for his commendable assistance with SWAT especially in running the new semi-automated calibration engine (SWAT-CUP). We also recognize Word bank through KAPLSMP for funding this research project.

**Table 1: Chronology of KAPSLMP PES project implementation in Sasumua watershed**

| Date | KAPSLMP activities | |
|---|---|---|
| | **Exploratory studies** | |
| 2005/2006 | PES project conceptualization | |
| 2008/2009 | PES feasibility study by ICRAF/PRESA | |
| 2009 | Situational analysis study | |
| 2011 | A study to evaluate the impacts of soil and water conservation practices on ecosystem services in Sasumua watershed, using SWAT model | |
| 2013 | The World Agroforestry Centre (ICRAF) in partnership with Jomo Kenyatta University of Agriculture and Technology (JKUAT) sought to explore the potential of using Payment for Ecosystems Services (PES) through its program, Pro-poor Rewards for Environmental Services in Africa (PRESA) | |
| 2014 | Policy and institutional analysis for PES | |
| 2014 | A study to assess water quality status of Sasumua Watershed (Baseline Results of key monitoring points) | |
| 2014 | A study to evaluate the potential for Payment for Ecosystem Services (PES) and policy implications in Sasumua watershed. | |
| **Actualizing 'theory into practice'** | | **M.sc research project activities** |
| February - March 2015 | Preliminary meetings with key stakeholders | M.sc Research Proposal and establishment of a representative watershed |
| March - May 2015 | Awareness creation on PES pilot project to potential small scale farmers, formation of CIGs and training | Baseline on water quality in the study site |
| June 2015 | Setting out - Establishment of individual Land Management Plans (LMPs) including field demonstrations | Baseline on SLM status in the study site |
| October 2015 | Training PES sub-committee on data collection | |
| October 2015 | Signing of PES contracts between the ES buyers and the sellers | |
| October - December 2015 | | Rainy season (Water quality sampling) |
| March 2016 | Auditing for PES rewards | |
| April 2016 | Rewarding of the leading farmers | |
| March - May 2016 | | Rainy season (Water quality sampling) |
| June 2016 | | Field study to establish SLM progress in the study site after one year of PES implementation |

**Table 2: Target SLM technologies under KAPSLMP PES project in Sasumua watershed**

| SLM Technology | Njabini MC | Kiburu MC | Githabai MC | Mukeu MC | TOTAL Target |
|---|---|---|---|---|---|
| Terrace (M) | 46,587 | 15,613 | 2,693 | 18,312 | 83,205 |
| Drainage channels(M) | 6,235 | 4,855 | 34,127 | 9,703 | 54,920 |
| Retention ditch (M) | 3,058 | 3,285 | 2,839 | 4,207 | 13,389 |
| Cut off Drains (M) | 10,861 | 350 | 16,933 | 1,761 | 29.905 |
| Grass strip (M) | 15,391 | 33,195 | 11,145 | 14,622 | 74,353 |
| Riverbank protection (M) | 2,130 | 79 | 0 | 957 | 3,166 |
| No. of grass splits | 206,301 | 159,756 | 178,513 | 126,763 | 671,333 |
| No. of trees (forest/fodder/fruit) | 13,823 | 25,511 | 20,229 | 12,221 | 71,784 |
| | | | | | n = 1,017 |

**Table 3: Land cover percentages in the study site**

|  | Sub-basin 1 | Sub-basin 2 | Sub-basin 3 |
|---|---|---|---|
| Forest | 351.52 Ha(96.7%) | 174.95 Ha (86.4%) | 6.03 Ha (14.4%) |
| Pasture | 0.58 Ha (0.2%) | 1.22 Ha (0.6%) | 2.28 Ha (5.4%) |
| Agriculture | 11.50 Ha (3.1%) | 26.25 Ha (13%) | 33.63 Ha (80.2%) |
| Total | 363.60 ha | 202.42 ha | 41.94 ha |

**Table 4: Land use in the representative sub-watershed**

| Land use | Area |
|---|---|
| Forest | 87.6 % |
| Pasture | 0.7 % |
| Agriculture | 11.7 % |

**Table 5: Model input data information**

| Data type | Source | Description |
|---|---|---|
| DEM | USGS website | 30 m resolution, U.S. Geological Survey (USGS) |
| Soils | SOTER | 1:1,000,000 Soil data was extracted from the Digital Soil and Terrain Database of East Africa (SOTER). |
| Land use | USGS website | Landcover maps were generated from 2016 landsat image obtained from USGS website using maximum likelihood method classification |
| Weather (1970-2014)[a] | Weather records for three stations in the watershed; Agricultural training centre (9036152), South Kinangop forest station (9036164) and Sasumua dam station (9036188) | Minimum and maximum daily temperature, daily precipitation, relative humidity, solar and wind |
| Rainfall[b] | Observed between January and May 2016 | Daily precipitation |
| Crop management | Field survey, (Mwangi et al., 2014) | Interviews with key informants and individual farmers on historical and current field crops and reviewing available literature. |
| Stream flow | Observed during March 18th – May 27th | Daily stream flow (m3/s) |
| Water quality | Observed during three rainy seasons March – May 2015 October – December 2015 March – May 2016 | Daily Total Suspended Solids (TSS) |

[a]Weather data between 1970 and 2014 was used in generating a weather generator data (WGN) for the study site using WGNmaker4 (Boisrame, 2016). This was particularly important in simulating the weather data that was not actually measured during the study period e.g solar, wind, relative humidity etc.

[b]The rainfall observed during the study period was used in SWAT calibration and in focusing the modelling to specific period of simulation (starting and ending dates) during which actual data collection on flow and Total suspended solids (TSS) was carried out.

**Table 6: Selected land management operations with their adjusted parameters to reflect best and worst scenarios in SWAT**

| Main SLM Technologies promoted under PES pilot project | Land Management selected for representation in SWAT | Adjusted Parameters in SWAT | Description | Parameter ranges | Adjusted to reflect Best scenario | Adjusted to reflect Worst scenario |
|---|---|---|---|---|---|---|
| FanyaJuu terraces and retention ditches | Terraces | TERR-P | To reflect reduced sediment losses | 0-1 | 0.18 | 1 |
| | | TERR-SL | To represent the minimum distance between the terraces in meters | 0-100 | 5 | 100 |
| | | TERR-CN | To account for increased infiltration | 20-100 | 76 | 99 |
| Contour farming | Contour planting | CONT_CN | To account for increased surface storage and infiltration | 20-100 | 76 | 99 |
| | | CONT_P | To account for decreased erosion | 0-1 | 0.18 | 1 |
| Grass strips | Filter strip | VFSRATIO | Ratio of field area to filter strip area | 0-300 | 30 | 300 |
| | | VFSCON | Fraction of the HRU which drains to the most concentrated 10% of filter strip | 0.25-0.75 | 0.75 | 0.25 |
| | | VFSCH | Fraction of the flow within the most concentrated 10% of the filter strip which is fully channelized | 0-100 | 5 | 100 |
| | | FILTERW | To account for increasing trapping efficiency of the filter strip | 0-100 | 5 | 0 |
| Planting of fruit/fodder trees along the contour (deep rooted) and planting crops (shallow rooted) in between the strips | Strip cropping | STRIP_N | To represent increased surface roughness in the direction of flow | 0.001-0.5 | 0.5 | 0.001 |
| | | STRIP_CN | To account for increased infiltration | 20-100 | 76 | 99 |
| | | STRIP_C | To reflect the average value of the multiple crops within the field | 0-1 | 0.4 | 0.4 |
| | | STRIP_P | To account for decreased erosion | 0-1 | 0.18 | 1 |

**Table 7: Progress of SLM adoption through the PES pilot project in the study site**

| SLM Technology | A
Baseline status | B
After 1 year | C
Baseline status
$= A/F$ | D
After 1 year
$= B/F$ | E
Change
$= \dfrac{B-A}{F}$ | F
Target |
|---|---|---|---|---|---|---|
| Terrace (M) | 820 | 1,561 | 9% | 18% | 8% | 8,820 |
| Retention ditch (M) | 120 | 551 | 4% | 18% | 14% | 3,120 |
| Grass strip (M) | 920 | 3725 | 10% | 42% | 32% | 8,820 |
| Riverbank protection (M) | 210 | 520 | 21% | 51% | 31% | 1,015 |
| No. of Napier splits | -- | 10865 | -- | 47% | -- | 22,925 |
| No. of Forest trees | -- | 183 | -- | 15% | -- | 1185 |
| No. of Fruit trees | -- | 1095 | -- | 96% | -- | 1145 |
| *Average SLM adoption* | | | *11%[a]* | *32%[a]* | *21%* | |

[a]Only Terraces, Retention ditches, Grass strips and Riverbank protection were considered (proxy indicators of SLM adoption) as they were easily measurable in baseline and the current status.

**Table 8: Model parameters considered in SWAT-CUP calibration**

| Parameter | Description | Minimum-maximum | Default values | Final calibrated values |
|---|---|---|---|---|
| **Parameters sensitive to Discharge** | | | | |
| CN2.mgt | Initial SCS runoff curve number for moisture condition II | 35 - 98 | 83 | 81.4 |
| ALPHA_BF.gw | Base flow alpha factor (1/days) | 0 - 1 | 0.048 | 0.6 |
| GW_DELAY.gw | Ground water delay time (days) | 0 - 500 | 31 | 19.9 |
| GWQMN.gw | Threshold depth of water in the shallow aquifer required for return flow to occur (mm $H_2O$) | 0 - 5000 | 1000 | 2809.3 |
| ESCO.hru | Soil evaporation compensation factor | 0 - 1 | 0.95 | 0.1 |
| EPCO.hru | Plant uptake compensation factor | 0 - 1 | 1 | 0.6 |
| SOL_K.sol | Saturated hydraulic conductivity (mm/hr) | 0 - 2000 | 65 | 252.7 |
| SOL_AWC.sol | Available water capacity of the soil layer (mm $H_2O$/mm soil) | 0 - 1 | 0.3 | 0.2 |
| SLSUBBSN.hru | Average slope length (m) | 10 - 150 | 60.98 | 74.8 |
| CH_K2.rte | Effective hydraulic conductivity in main channel alluvium (mm/hr) | -0.01 - 500 | 0 | 27.9 |
| OV_N.hru | Manning's N value overland flow | 0.01 - 30 | 0.14 | 29.9 |
| HRU_SLP.hru | Average slope steepness (m/m) | 0 – 0.6 | 0.0766 | 0.5 |
| **Parameters sensitive to sediments** | | | | |
| SPEXP.bsn | Channel re-entrained exponent parameter | 1 - 1.5 | 1 | 1.2 |
| SPCON.bsn | Channel re-entrained linear parameter | 0.0001 - 0.01 | 0.0001 | 0.00237 |
| CH_EROD.rte | Channel erodability factor | 0 - 1 | 0 | 0.2 |
| CH_COV.rte | Channel cover factor | -0.001 - 1 | 0 | 0.6 |
| USLE_P.mgt | Support practice factor | 0 - 1 | 1 | 0.8 |

**Table 9: Rainfall characteristics for March – May 2015 and March – May 2016**

| Seasonal Totals | MAM 2015 (mm) | MAM 2016 (mm) |
|---|---|---|
| | 811.7 | 912.6 |
| Monthly Totals | MAM 2015 (mm) | MAM 2016 (mm) |
| March | 93.8 | 52.6 |
| April | 384.1 | 412.1 |
| May | 333.8 | 447.9 |
| | MAM 2015 (mm) | MAM 2016 (mm) |
| Number of rain Days | 46 | 58 |
| Average (mm) per day | 8.82 | 9.92 |
| Number of rainfall days exceeding 30mm per day | 7 | 8 |

*t-Test: Two-Sample Assuming Unequal Variances*

| | MAM 2015 (mm) | MAM 2016 (mm) |
|---|---|---|
| Mean | 8.822826087 | 9.919565217 |
| Variance | 194.8576051 | 217.1771954 |
| Observations | 92 | 92 |
| Hypothesized Mean Difference | 0 | |
| df | 181 | |
| t Stat | -0.518239252 | |
| P(T<=t) one-tail | 0.302462135 | |
| t Critical one-tail | 1.653315758 | |
| P(T<=t) two-tail | 0.60492427 | |
| t Critical two-tail | 1.973157001 | |

**Table 10: Test for significant differences for TSS (March - May 2015 and March - May 2016)**

*t-Test: Two-Sample Assuming Unequal Variances*

| | TSS (MAM2015) | MAM 2016 |
|---|---|---|
| Mean | 71.0501356 | 42.73239437 |
| Variance | 1273.523047 | 577.484507 |
| Observations | 31 | 71 |
| Hypothesized Mean Difference | 0 | |
| df | 42 | |
| t Stat | 4.036527381 | |
| P(T<=t) one-tail | 0.000112488 | |
| t Critical one-tail | 2.418470354 | |
| P(T<=t) two-tail | 0.000224976 | |
| t Critical two-tail | 2.69806618 | |

[Figure]

**Figure 1: The representative sub-watershed (headwater)**

[Figure]

**Figure 2: Simulated impact of PES scheme on water quality**

[Figure]

**Figure 3: Sasumua Dam Rain gauge station**

[Figure]

**Figure 4: Rainfall series for both March-May 2015 and March-May 2016**

[Figure]

**Figure 5: Progress of SLM adoption after one year of PES implementation**

[Figure]

5    **Figure 6: The relationship between Lovibond TSS and gravimetric TSS**

[Figure]

**Figure 7: TSS results of the three points (A, B & E) during the two rainy seasons of March-May 2015 and March-May 2016**

[Figure]

**Figure 8: Flow calibration and validation**

[Figure]

**Figure 9: Sedimentations calibration and validation**